# A case study for a psychographic-behavioral segmentation approach for targeted demand generation in voluntary medical male circumcision

Sema K Sgaier[1,2,3]*, Maria Eletskaya[4], Elisabeth Engl[1], Owen Mugurungi[5], Bushimbwa Tambatamba[6], Gertrude Ncube[5], Sinokuthemba Xaba[5], Alice Nanga[7], Svetlana Gogolina[7], Patrick Odawo[8], Sehlulekile Gumede-Moyo[7,9], Steve Kretschmer[1]

[1]Surgo Foundation, Seattle, United States; [2]Department of Global Health and Population, Harvard T.H. Chan School of Public Health, Boston, United States; [3]Department of Global Health, University of Washington, Seattle, United States; [4]Cello Health Insight, London, United Kingdom; [5]Ministry of Health and Child Care, Harare, Zimbabwe; [6]Mother and Child Health, Ministry of Community Development, Lusaka, Zambia; [7]Ipsos Healthcare, London, United Kingdom; [8]Ahimsa Group LLC, Nairobi, Kenya; [9]Faculty of Epidemiology and Population Health, London School of Hygiene and Tropical Medicine, London, United Kingdom

*For correspondence:
semasgaier@surgo-foundation.org

Competing interests: The authors declare that no competing interests exist.

**Abstract** Public health programs are starting to recognize the need to move beyond a one-size-fits-all approach in demand generation, and instead tailor interventions to the heterogeneity underlying human decision making. Currently, however, there is a lack of methods to enable such targeting. We describe a novel hybrid behavioral-psychographic segmentation approach to segment stakeholders on potential barriers to a target behavior. We then apply the method in a case study of demand generation for voluntary medical male circumcision (VMMC) among 15–29 year-old males in Zambia and Zimbabwe. Canonical correlations and hierarchical clustering techniques were applied on representative samples of men in each country who were differentiated by their underlying reasons for their propensity to get circumcised. We characterized six distinct segments of men in Zimbabwe, and seven segments in Zambia, according to their needs, perceptions, attitudes and behaviors towards VMMC, thus highlighting distinct reasons for a failure to engage in the desired behavior.
DOI: https://doi.org/10.7554/eLife.25923.001

## Introduction

Voluntary Medical Male Circumcision (VMMC) is a critical strategy for HIV prevention in countries with high HIV and low circumcision prevalence (*Auvert et al., 2005*; *Bailey et al., 2007*; *Gray et al., 2007*; *World Health Organization and Joint United Nations Programme on HIV/AIDS, 2007*; *Njeuhmeli et al., 2011*; *World Health Organization and UNAIDS, 2011*). The VMMC program has been scaling up rapidly across 14 eastern and southern African countries and close to 12 million circumcisions, against the 20 million target among 15–49 year old males, have been achieved to date (*World Health Organization, 2016*). However, given that resources for VMMC scale-up are finite, achieving the desired impact of VMMC on reducing HIV incidence will require prioritizing sub-

**eLife digest** Companies invest a significant amount of time and money into market research that helps them to understand the behaviors, beliefs and motivations of their potential customers. By then "segmenting" people into groups according to these characteristics, marketing messages can be produced that target specific groups more effectively.

Most public health efforts are either mass communication campaigns or target particular age groups. However, some public health organizations are starting to study whether the segmenting tactics used by companies could also help to promote healthy behaviors. For example, male circumcision has been shown to reduce the transmission of HIV in Africa. Identifying the beliefs, emotions, motivations or other barriers that stop men from getting circumcised and then targeting specific messages to different groups could help to increase the number of men who opt for circumcision.

Sgaier et al. now present evidence that suggests that segmentation could help to promote circumcision in Zimbabwe and Zambia. 4,000 men from these countries answered a survey that had been designed based on previous research that investigated how men make the decision whether to be circumcised. Analyzing the results using k-means clustering, a machine learning algorithm, enabled Sgaier et al. to identify six distinct segments in the men from Zimbabwe and seven in the men from Zambia. Further analyses found that the risk of contracting HIV also varied by segment.

Sgaier et al. then demonstrated that field workers could use a series of questions to allocate men to each of the groups with an accuracy of over 60%. The segmentation method therefore looks like a promising tool that could be applied to a wide range of public health campaigns. As well as targeting specific groups of people with messaging that resonates specifically with them, segmentation could also highlight those people who are likely to be most easily convinced by a particular health intervention. More research is now needed to improve the usability of the tools that field workers can use to segment their audience.

DOI: https://doi.org/10.7554/eLife.25923.002

populations of males and generating demand for VMMC amongst them (*Sgaier et al., 2015*; *Sgaier et al., 2014*).

There is a need for a greater evidence base, including the use of new approaches adapted from the private sector, to inform the design of demand generation approaches for VMMC (*Sgaier et al., 2015*). In particular, taking a 'market segmentation' approach has been highlighted as an important strategy to implement and enable more efficient use of resources. With 'market segmentation', the target population is sub-divided into groups where members of the group share elements in common (*Thomas, 2016*; *Wedel and Kamakura, 2006*). Segmenting the target populations along demographics, such as age or geographic location, has been commonly used in public health. In the private sector, the use of behavioral or psychographic segments has been prevalent (*Bhatnagar and Ghose, 2004*; *Wade and Eagles, 2003*; *Desarbo et al., 1995*; *Gloy and Akridge, 1999*). These segments are constructed based on factors including shared attitudes, values, emotions, perceptions, beliefs and behaviors. Given that these factors play a critical role in driving decisions, often more so than age or other demographic factors (*Carpenter, 2010*; *Velicer et al., 2007*), segmenting the population along psychographic and behavioral parameters should provide a more powerful tool to target them effectively for behavior change. The application of segmentation has been such a successful strategy to marketing of products and services to consumers that it is now often executed to inform the design of most marketing campaigns in the private sector. We are not aware of any public health or development program that demonstrated the application of behavioral psychographic segmentation. On the other hand, many academic models of behavior, such as the Health Belief Model, Self-determination Theory, and the Reasoned Action Approach, have used differences between individuals in behavioral variables such as (risk) perception, beliefs, and motivations to explain why a behavior has or has not occurred (*Carpenter, 2010*; *Armitage and Conner, 2001*; *Fishbein and Ajzen, 2010*; *Ryan and Deci, 2000*). However, segmentation techniques are not well-established in the application of these theories to design interventions.

We designed and implemented a hybrid behavioral-psychographic segmentation study in Zambia and Zimbabwe. This quantitative study was conducted among males 15–29 years old, given previous evidence that identified this as the most efficient and impactful age for the VMMC programs in both countries to target (*Awad et al., 2015a*; *Awad et al., 2015b*). In an age-structured mathematical model, Awad et al. (*Awad et al., 2015a*; *Awad et al., 2015b*) assessed the impact of prioritizing different age groups for VMMC in Zimbabwe (*Awad et al., 2015a*) and Zambia (*Awad et al., 2015b*). The model took multiple factors into account, including VMMC effectiveness, cost-effectiveness, reduction in HIV incidence, program cost, and programmatic feasibility. Overall, the analysis found that targeting young males, especially the 15–29 age group, combined some of the largest reductions in HIV incidence with some of the highest program efficiency gains. Therefore, we focused our study on this age group. Both psychographic and behavioral factors were investigated to reveal specific groups of men based on their needs, perceptions, attitudes and behaviors towards VMMC.

This research was built on findings from an integrated qualitative study that combined consumer journey mapping, a market research approach that maps the context and sequence of experiences and interactions along the path to decisions from the perspective of the individual, and a behavioral economics game that simulates the real world context of participants to identify context, emotions and mental models that underlie behaviors (Eletskaya M, Sgaier SK, Kretschmer S, Prasad R, Mulhausen J, Vaish A. 2017. Employing consumer journey research and behavioral science to understand decision making for Voluntary Medical Male Circumcision in Zimbabwe and Zambia, in preparation). In both countries, this integrated qualitative study identified the men's journey to getting circumcised, the intention-to-action gap that stalled the journey of many to undergoing circumcision, and the underlying factors that inhibited or facilitated the progression of different men to complete their journey.

We also explored how the different segments differed in their perceived and actual risk towards HIV/AIDS infection. Finally, we developed a typing algorithm to help programs classify men into the different segments and profile men on risk. Collectively, the insights and tools enabled the development of more effective demand generation activities targeted to specific segments.

To our knowledge, this is the first study that uses a hybrid behavioral-psychographic segmentation approach in a public health program at national scale. Our approach, analysis and findings will not only help the national VMMC programs in Zambia and Zimbabwe, but also lay the groundwork for segmentation based on behavioral variables to be used in other development programs.

## Results

### Population characteristics

*Figure 1—figure supplement 1* shows the demographic and cultural characteristics of the population sample in Zambia and Zimbabwe. *Figure 1—figure supplement 2* characterizes the social acceptability of VMMC, as well as the perceived risk of HIV and other sexually-transmitted infections (STIs) in both countries' samples. The two populations are broadly comparable on the measured parameters, such as age, education level, religion, working status, the perception whether many men had already been circumcised in the community, reasons for circumcision (for circumcised men only), and the perceived risk for HIV/STIs. The starkest difference was that in Zambia, a much greater share of the population was only educated to primary-school level, and a smaller percentage was employed than in Zimbabwe.

### Identification of segments

In order to explain the differences between segments, the results of canonical correlations were analyzed and five key roots influencing intention to choose VMMC were identified. Three of them were common in both countries, while two had some country specifics. The last two roots in Zambia were eliminated from segments profiling due to relatively low contribution into variance between segments. Motivation to go for VMMC, which includes intention to go/advocate for circumcision combined with positive attitudes and beliefs about VMMC, was the most important factor (similar in both countries). It was followed by rejection due to cognitive dissonance in Zimbabwe and control over cognitive dissonance in Zambia, which explains rejection of VMMC and self-efficacy in going for the procedure (this factor combines, on the one hand, high concern about HIV, and, on the other

hand, doubts in VMMC efficacy in regard to protection from HIV along with signs of cognitive disso-nance, such as interest to circumcision along with hesitation and uncertainty in its need). These two factors were followed by perceived lack of ability (for both countries it includes perceived level of knowledge about VMMC and desire to have more information/hesitation due to lack of information about the procedure) and acceptance of social support (in Zimbabwe)/perceived self-efficacy against social pressure (in Zambia). The latter includes self-efficacy to go for circumcision even if it is not accepted by people around (in Zambia) or readiness to provide social support for others and accept it from others (in Zimbabwe) combined with perceived level of social support (from family and wider community). The last factor was personal constraints (such as fears of pain and embarrassment dur-ing the procedure and concerns about the surgery and healing process) which also was common in both countries. Based on these factors, six segments in Zimbabwe and seven segments in Zambia were identified (*Table 1A and B*). Each segment was profiled overall based on three levels (low, medium, or high expression) of each factor, explaining the characteristics of and differences among the segments (*Table 1A and B*). For instance, a segment in Zimbabwe characterized by strong levels of motivation, neutral levels of rejection due to cognitive dissonance, an average perceived lack of ability, high acceptance of social support, and moderate levels of fear of the procedure was called Enthusiasts (see *Table 1A and B* for a characterization of all segments, and *Figure 1* for an overview of the constructs investigated). There was almost an even distribution of the population across the different segments in each country (*Figure 2A and B*)

The segments can be prioritized by the program for targeting using a number of criteria. Below we illustrate two: ease of converting to higher levels of VMMC uptake and estimated behavioral risk for acquiring HIV/AIDS.

## Composition of the different segments, potential opportunity, and ease of conversion for each segment

The segments are strongly differentiated in terms of the proportion of men who have already been circumcised, which shows that attitudes, motivations and belief correlate with circumcision behavior (*Table 2*). Most of the men in the Friends-driven Hesitant (86%), Scared Rejecters (90%) and Indiffer-ent Resisters (94%) in Zambia and Neophytes (94%), Rejecters (97%), Embarrassed Rejecters (67%), and Highly Resistant (99%) in Zimbabwe had not been circumcised. The highest levels of circumci-sion are found amongst the Self-reliant Believers (71%) and Traditional Believers (71%) in Zambia and Champions (76%) in Zimbabwe. However, given that the sizes of the segments are different, the relative distribution of uncircumcised men by segment is different (*Figure 2A–D*). Whilst the Cham-pions represent 17% of uncircumcised men in Zimbabwe, only 6% of uncircumcised men are found in this segment and therefore represent low potential for targeting. Similarly, in Zambia, 19% of the men are Self-reliant Believers, but only 9% of uncircumcised men are found within this segment.

Given that males are at different stages of their journey towards VMMC (Eletskaya M et al., in preparation), targeting uncircumcised males who are in the 'committed' stage could be one strategy that the program could employ to achieve its targets (*Figure 2—figure supplement 1*). We see con-siderable differences among segments on proportion of males who are uncircumcised and commit-ted (*Table 2*). Proportions of uncircumcised, committed males to VMMC also vary from segment to segment, providing the program for opportunities to target the 'low-hanging fruit' (*Figure 2E and F*).

In Zimbabwe, the Enthusiasts represent the biggest opportunity for the program. Slightly more than half of the men in this segment are uncircumcised, but at the same time are strongly committed to circumcision (*Table 2*). They represent 38% of the males in the population who are uncircumcised and committed to choose VMMC (*Figure 2E*). These men want to get circumcised, but need some additional support to cope with their fears and cognitive dissonance. These men also have the high-est potential to advocate for VMMC once circumcised (*Figure 2—figure supplement 1*). The second level of potential is within the Embarrassed Rejecters and Neophytes (*Figure 2E*). Neophytes can be converted more easily by addressing their knowledge gaps and dissonance. Embarrassed Rejecters will benefit from having more circumcised men around them, as they are highly influenced by a sense of social inclusion and acceptance. While the Highly Resistant represent 21% of the uncircumcised males, they are very difficult to convert (only 3% of the uncircumcised committed males are within this segment). By just targeting three segments (Enthusiasts, Embarrassed Rejecters and Neophytes), Zimbabwe can circumcise close to 50% of the uncircumcised men in the population.

**Table 1.** (A) Factors deriving segments and segment profile summaries (Zimbabwe). (B) factors deriving segments and segment profile summaries (Zambia).

**Table 1A – Factors deriving segments and segment profile summaries (Zimbabwe)**

| Country | Segment | Key factors defining segment profiles | | | | | Summary of differences among segments |
|---|---|---|---|---|---|---|---|
| | | Motivation/ need for VMMC | Rejection due to cognitive dissonance | Perceived lack of ability | Acceptance of social support | Personal constraints | |
| Zimbabwe | Enthusiasts | Strong motivation | Neutral | Average ability | Highly socially driven | Some fears | Believe in all benefits of VMMC (including benefits related to sexual life); emotionally associate VMMC with a sense of achievement; relatively high level of risky sexual behavior; very socially driven and supported by social environment; require support to overcome some fears and cognitive dissonance, and strengthen ability to go for VMMC. |
| | Champions | Strong motivation | No rejection | Strong ability | Highly independent | Some fears | Have positive attitudes to VMMC; believe in benefits; much more socially independent (going for VMMC is their own decision, not driven by social environment); feel strong ability to go for VMMC; despite presence of some fears, don't experience serious cognitive dissonance. |
| | Neophytes | Neutral motivation | Strong rejection | Lack of ability | Highly independent | Some fears | More ambivalent attitude to VMMC (have not decided yet whether they need it or not); quite low level of risky sexual behavior; feel lack of control and rejection due to cognitive dissonance; feel lack of knowledge about VMMC (need information); not socially driven. |
| | Scared Rejecters | Neutral motivation | Strong rejection | Strong ability | Highly independent | Strong fears | Weak motivation due to a number of fears; very worried about contraction of infections and need additional sense of protection, but are not able to go for VMMC (due to fears of complications, pain, surgery, healing process, etc.); feel strong ability to go for VMMC (no need in additional information); not socially driven. |
| | Embarrassed Rejecters | Weak motivation | No rejection | Average ability | Highly socially driven | Strong fears | Weakly motivated to go for VMMC; mostly are not concerned about HIV/STI contraction; have mostly negative beliefs about VMMC; due to absence of motivation do not experience cognitive dissonance; have some positive believes (especially, believe in hygiene), but largely don't consider VMMC for themselves; have fears and concerns; highly socially driven; have mostly no social support for VMMC. |
| | Highly Resistant | Weak motivation | Strong rejection | Strong ability | Highly socially driven | No fears | Weak motivation, rejection of VMMC; mostly negative beliefs about VMMC; relatively higher risk of HIV/STI contraction; however, level of concern about HIV/STIs contraction is low; are not open to information and feel that they know all they need about VMMC; claim absence of fear; very socially driven; mostly highly rejecting VMMC social environment. |

**Table 1B – Factors deriving segments and segment profile summaries (Zambia)**

| Country | Segment | Key factors defining segment profiles | | | | | Summary of differences among segments |
|---|---|---|---|---|---|---|---|
| | | Motivation/ need for VMMC | Control over cognitive dissonance | Perceived lack of ability | Self-efficacy against social pressure | Personal constraints | |
| Zambia | Socially-supported believers | Strong motivation | Strong confidence | Average ability | Fully independent | Strong fears | Strong motivation for VMMC; high level of concern about contraction of HIV/STIs; believe in majority of benefits, emotionally associate VMMC with sense of belonging feel that they are independent from social environment in their decision to go for VMMC; but are very actively supported by all people around; control cognitive dissonance; have some minor fears. |
| | Self-reliant believers | Strong motivation | Average confidence | Strong ability | Somewhat socially driven | Strong fears | Strong motivation; believe in the benefits of VMMC, tend to value the benefits for sexual relationships; emotionally associate VMMC with a feeling of closeness to their partner; less socially supported, which makes them slightly less confident in themselves; feel some cognitive dissonance, but presence of fears doesn't make them doubt in necessity of VMMC; feel strong ability to go for VMMC, don't require additional information. |
| | Knowledgeable Hesitant | Neutral | Strong confidence | Lack of ability | Somewhat socially driven | No fears | Somewhat motivated to go for VMMC, mostly because of HIV/STI protection; also value benefits for sexual life, considering that circumcised men are more desired by women; riskier sexual behavior; quite strong concerns about possible negative consequences of VMMC, which make them hesitate (e.g. safety of the procedure; loss of sensitivity, increase of promiscuity). |
| | Friends-Driven Hesitant | Neutral | Lack of confidence | Lack of ability | Somewhat socially driven | No fears | Ambivalent attitude toward circumcision: not completely rejecting circumcision, but also don't have strong motivation; less risky sexually, mostly focused on hygiene benefit; social environment also has two directions (supporting and inhibiting); lack of assurance in the need of VMMC; however, don't have any serious personal constraints. |
| | Scared Rejecters | Weak motivation | Average confidence | Lack of ability | Fully independent | Strong fears | Mostly negative attitude to VMMC; negative beliefs in possible complications, doubts in safety of the procedure; experience cognitive dissonance; don't completely reject the benefits of VMMC, but the protection of infections is less relevant for them; mainly believe in the hygiene benefit and help for sons to be circumcised. |
| | Indifferent Rejecters | Weak motivation | Strong confidence | Strong ability | Somewhat socially driven | Some fears | The least educated and least knowledgeable of VMMC; are generally not concerned about HIV and other infections; attitude to VMMC is ambivalent; relatively socially driven; not strongly supported by social environment. |
| | Traditional Believers | Neutral | Strong confidence | Lack of ability | Very socially driven | Some fears | Mostly circumcised or have a high level of commitment to VMMC; however rather poor knowledge about the benefits and belief in them; the proportion of men circumcised for religious/traditional reasons is the highest; the key benefit of circumcision is maintenance of tradition in the family, helping sons to get circumcised. |

The table summarizes the differences among segments based on the key factors identified via canonical correlations analysis. Zambia, 5 key factors, 7 segments; Zimbabwe, 5 key factors, 6 segments.
DOI: https://doi.org/10.7554/eLife.25923.008

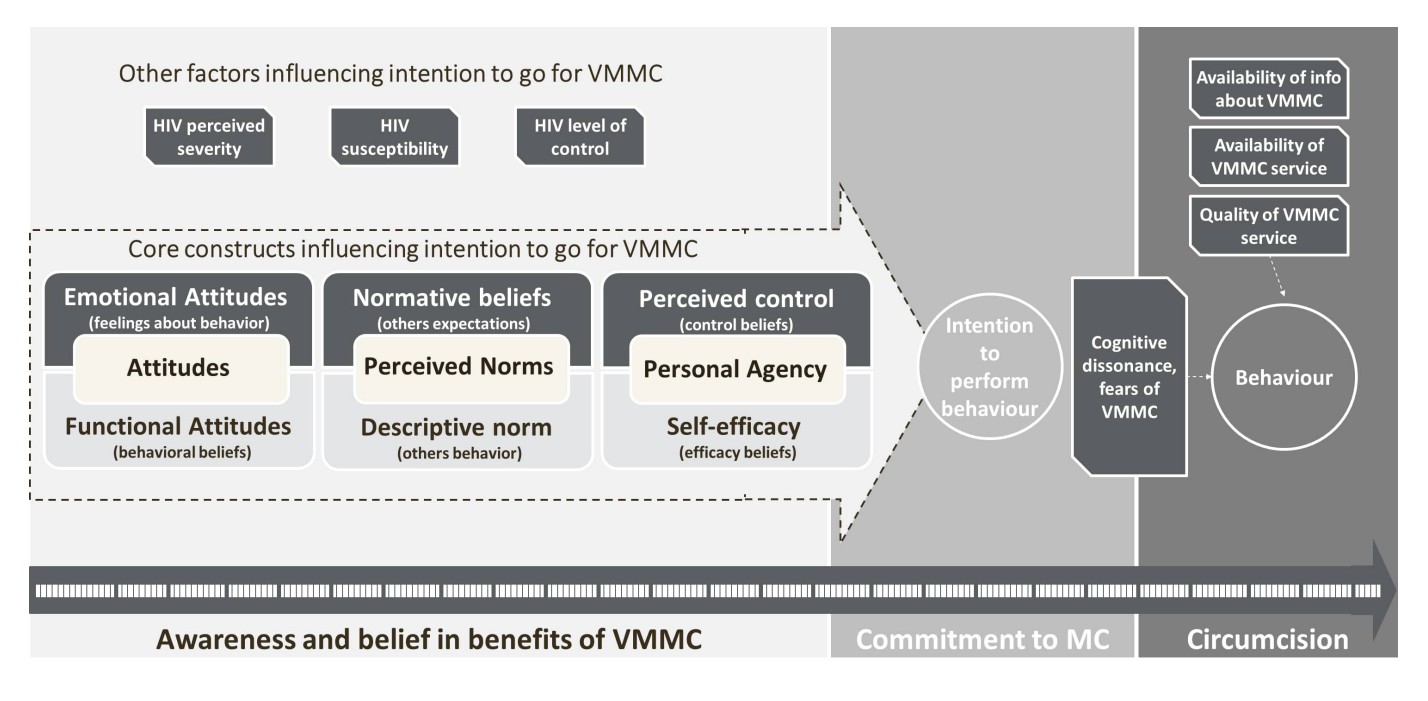

**Figure 1.** Segmentation questionnaire design construct.

DOI: https://doi.org/10.7554/eLife.25923.003

The following figure supplements are available for figure 1:

**Figure supplement 1.** Demographic and cultural characteristics of the sample population in Zambia and Zimbabwe.

DOI: https://doi.org/10.7554/eLife.25923.004

**Figure supplement 2.** Social acceptability of VMMC and perceived risk of HIV/STIs in the sample population in Zambia and Zimbabwe.

DOI: https://doi.org/10.7554/eLife.25923.005

The Friends-driven Hesitants and Enthusiasts represent the biggest opportunity in Zambia. Both have a large proportion of uncircumcised and committed men (*Table 2*). While the largest proportion of uncircumcised men in Zambia are within Indifferent Resistants, there are few who are committed. They require a lot of education and support from their social networks. The program could deprioritize the Traditional Believers as most of them will go for circumcision for traditional reasons eventually, anyway. The remaining segments are more or less the same in terms of ease of conversion and targeting each segment will require addressing specific needs (for example the safety concern of 'Knowledgeable Hesitants' or strong fear of surgical procedures of Scared Rejecters).

## Segment-specific risk for HIV/AIDS

On the other hand, the program may want to prioritize those segments that are at highest risk for acquiring HIV to have a more effective impact on the epidemic. In this case, the priority of segment targeting will change, as the estimated risk of acquiring HIV, as measured by the index we created, does not align with the likelihood to already be circumcised nor intent to get circumcised, by segment.

In Zimbabwe, the Highly Resistant segment is least likely to be circumcised or intend to get circumcised across all six segments in the country. However, the Highly Resistant segment is assessed as most at risk of acquiring HIV based on sexual behaviour, with 78% having moderate or high risk on the estimated risk index (see *Figure 3*). In Zimbabwe, then, if prioritization of segments is made by risk for acquiring HIV, the priorities would be the following: 1) Highly Resistant, 2) Embarrassed Rejecters (73% moderate to high risk), 3) Scared Rejecters (66% moderate to high risk), and then Neophytes, Enthusiasts and Champions (57%, 57% and 56% respectively).

In Zambia, estimated risk is less differentiated across the segments overall, with the priority being the following: (1) Scared Rejecters (67% moderate to high risk), (2) Indifferent Resistants,

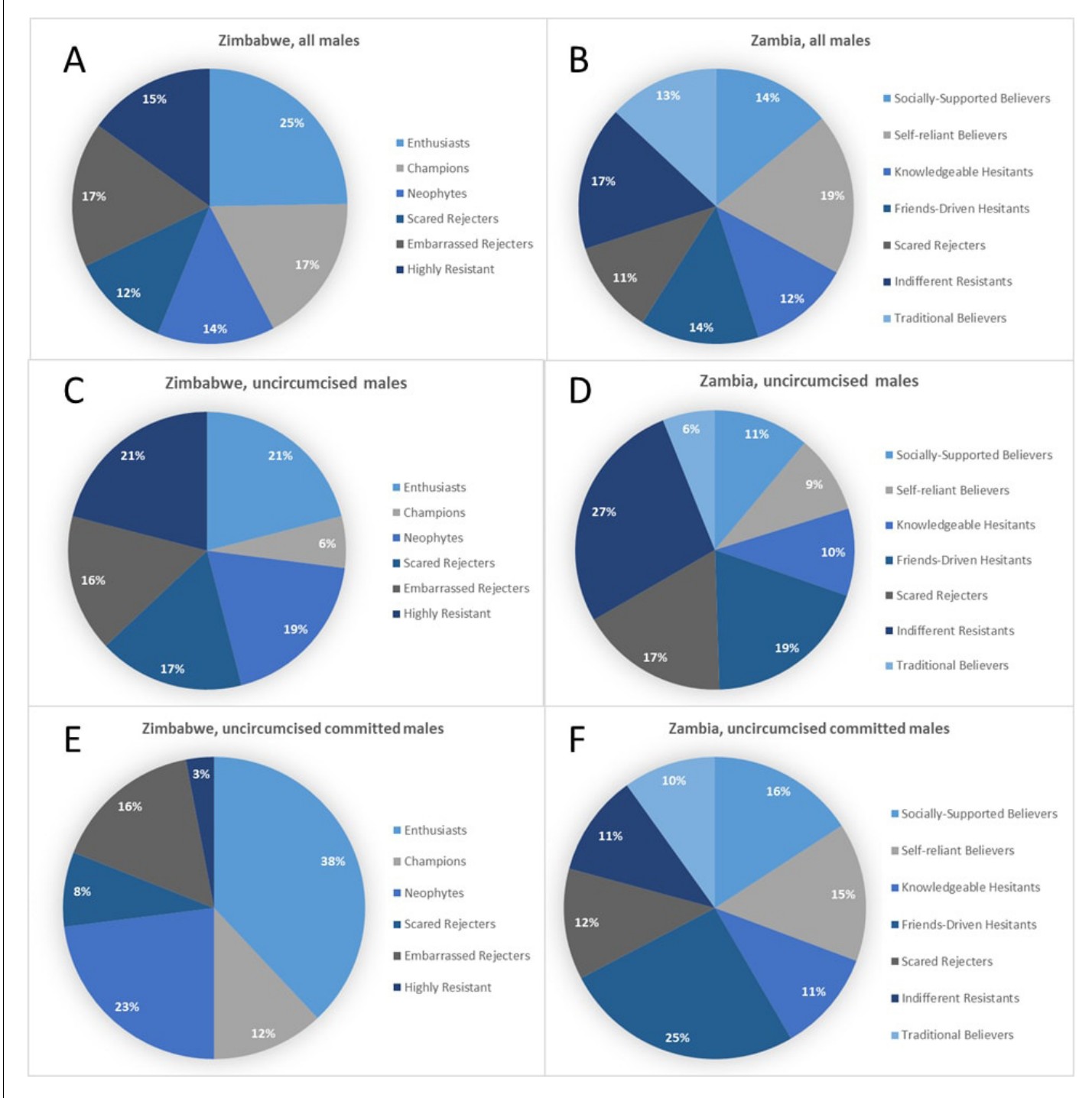

**Figure 2.** Distribution of males by segment.

DOI: https://doi.org/10.7554/eLife.25923.006

The following figure supplement is available for figure 2:

**Figure supplement 1.** Distribution of men in each stage of the decision-making journey within each segment.

DOI: https://doi.org/10.7554/eLife.25923.007

**Table 2.** Circumcision levels and commitment for MC, by segment.

| Country | Segment | All men in segment* | | Uncircumcised men in segment[†] | |
|---|---|---|---|---|---|
| | | Circumcised % (n) | Uncircumcised % (n) | Not committed % (n) | Committed % (n) |
| Zimbabwe | Enthusiasts | 42.6 (211) | 57.4 (284) | 15.5 (44) | 84.5 (240) |
| | Champions | 76.2 (269) | 23.8 (84) | 8.3 (7) | 91.7 (77) |
| | Neophytes | 6.1 (17) | 93.9 (260) | 43.1 (112) | 56.9 (148) |
| | Scared rejecters | 2.6 (6) | 97.4 (228) | 78.1 (178) | 21.9 (50) |
| | Embarrassed rejecters | 32.7 (112) | 67.3 (230) | 56.1 (129) | 43.9 (101) |
| | Highly resistant | 0.7 (2) | 99.3 (298) | 92.6 (276) | 7.4 (22) |
| Zambia | Socially-supported believers | 56.1 (160) | 43.9 (125) | 20.0 (25) | 80.0 (100) |
| | Self-reliant believers | 71.2 (272) | 28.8 (110) | 14.5 (16) | 85.5 (94) |
| | Knowledgeable hesitant | 49.8 (119) | 50.2 (120) | 41.7 (50) | 58.3 (70) |
| | Friends-driven hesitant | 14.1 (38) | 85.9 (231) | 29.0 (67) | 71.0 (164) |
| | Scared rejecters | 9.7 (22) | 90.3 (204) | 62.7 (128) | 37.3 (76) |
| | Indifferent resisters | 5.5 (19) | 94.5 (325) | 79.4 (258) | 20.6 (67) |
| | Traditional believers | 70.6 (180) | 29.4 (75) | 16.0 (12) | 84.0 (63) |

*No. of circumcised OR uncircumcised men in segment/no. total men in segment; *Zambia, N = 2000; Zimbabwe, N = 2001*

[†]uncircumcised committed OR not-committed men in segment/all uncircumcised men in segment; *Zimbabwe, N = 1384; Zambia, N = 1189*

DOI: https://doi.org/10.7554/eLife.25923.009

Knowledgeable Hesitants and Traditional Believers (65%, 64% and 64% respectively), (3) Self-reliant Believers and Friends-driven Hesitants (62% and 63% respectively) and then 4) Socially-supported Believers (57%) (see *Figure 3*).

Further assessing the segments based on their self-perceived risk is additionally helpful in understanding their approach to VMMC. For example, when evaluating the perceived vs. estimated risk profiles in Zambia, segments having a higher self-perception of being at risk of HIV infection already have higher rates of circumcision. An exception are Traditional Believers, who circumcise for traditional reasons rather than for health reasons; interestingly, their perception vs. estimated risk profile is like that of the Indifferent Resistants (see *Figure 3*). In contrast, in Zimbabwe, the perception vs. risk profiles do not align with circumcision rates as clearly. Neophytes, Enthusiasts and Embarrassed Rejecters have the highest rates of moderate and high perceived risk (76%, 73% and 71% respectively), but Champions have by far the highest rate of circumcisions.

## Segment identification algorithm

The segment identification algorithm for Zimbabwe resulted in 9 rating questions organized in a decision tree. In Zimbabwe, any given man (aged 15–29 years) needs to answer only 2, 3 or 4 of the questions, based on his path through the decision tree, to be classified into his appropriate segment (*Figure 4*). In Zambia, the algorithm similarly resulted in 8 questions, of which only 2 to 4 need to be asked to classify a man into his segment (Figure 4—figure supplement 1).

## Discussion

Adapting the use of 'market segmentation' from the private sector, this research demonstrated the application of a hybrid behavioral-psychographic segmentation for men to investigate types of obstacles to demand for VMMC in Zambia and Zimbabwe. Importantly, the research design was built using prior evidence that identified underlying factors – beliefs, barriers, influences – that inhibit or facilitate men to mentally commit to and then take action to getting circumcised (*Montaño et al., 2014*; *Price et al., 2014*). Characterizing males aged 15–29, the resulting segmentation solutions in each country identified segments which are strongly differentiated in their levels of circumcised men, levels of commitment to VMMC among uncircumcised men and, crucially, the underlying factors, e.

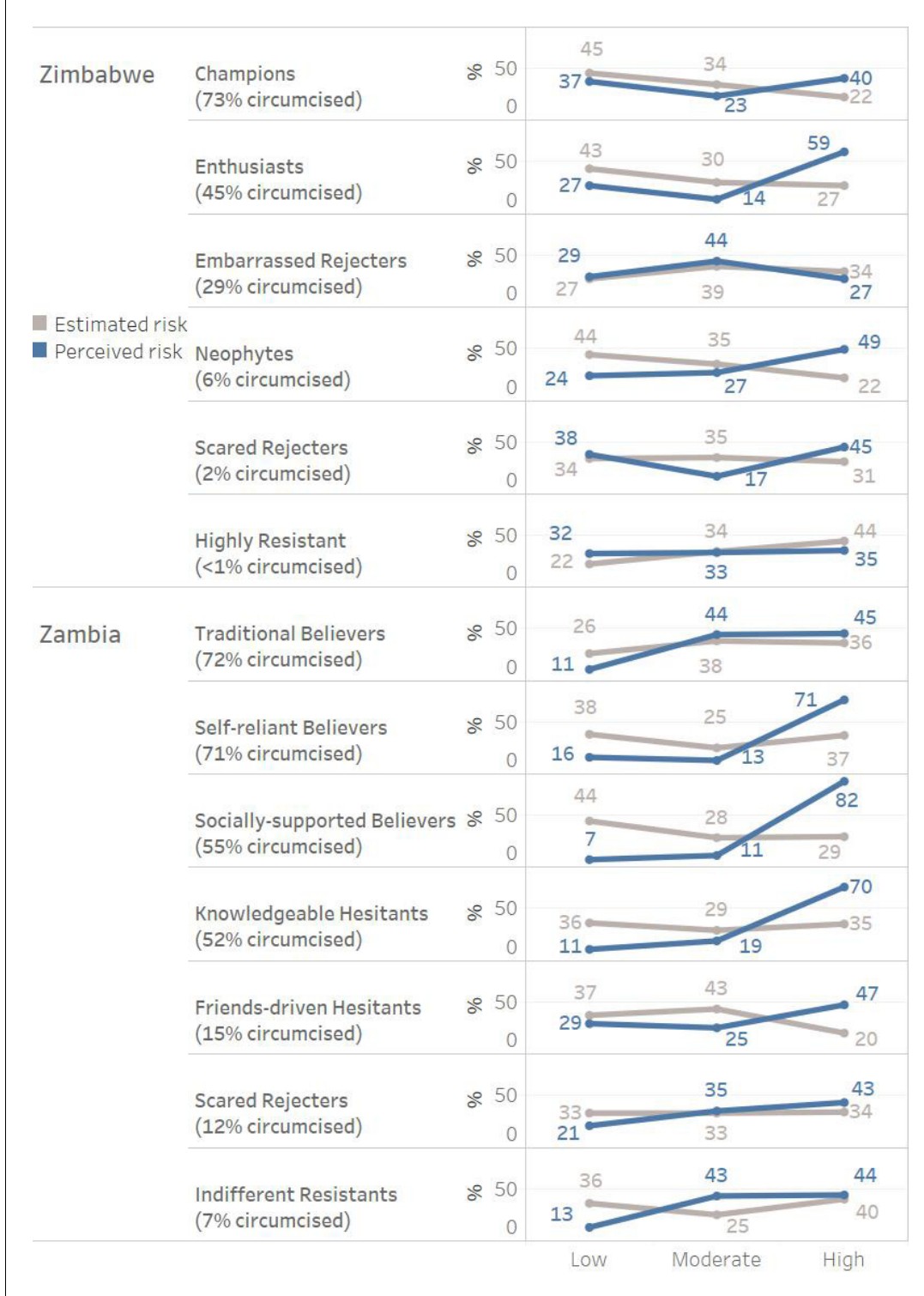

**Figure 3.** Estimated vs. perceived HIV infection risk by segment.
DOI: https://doi.org/10.7554/eLife.25923.010

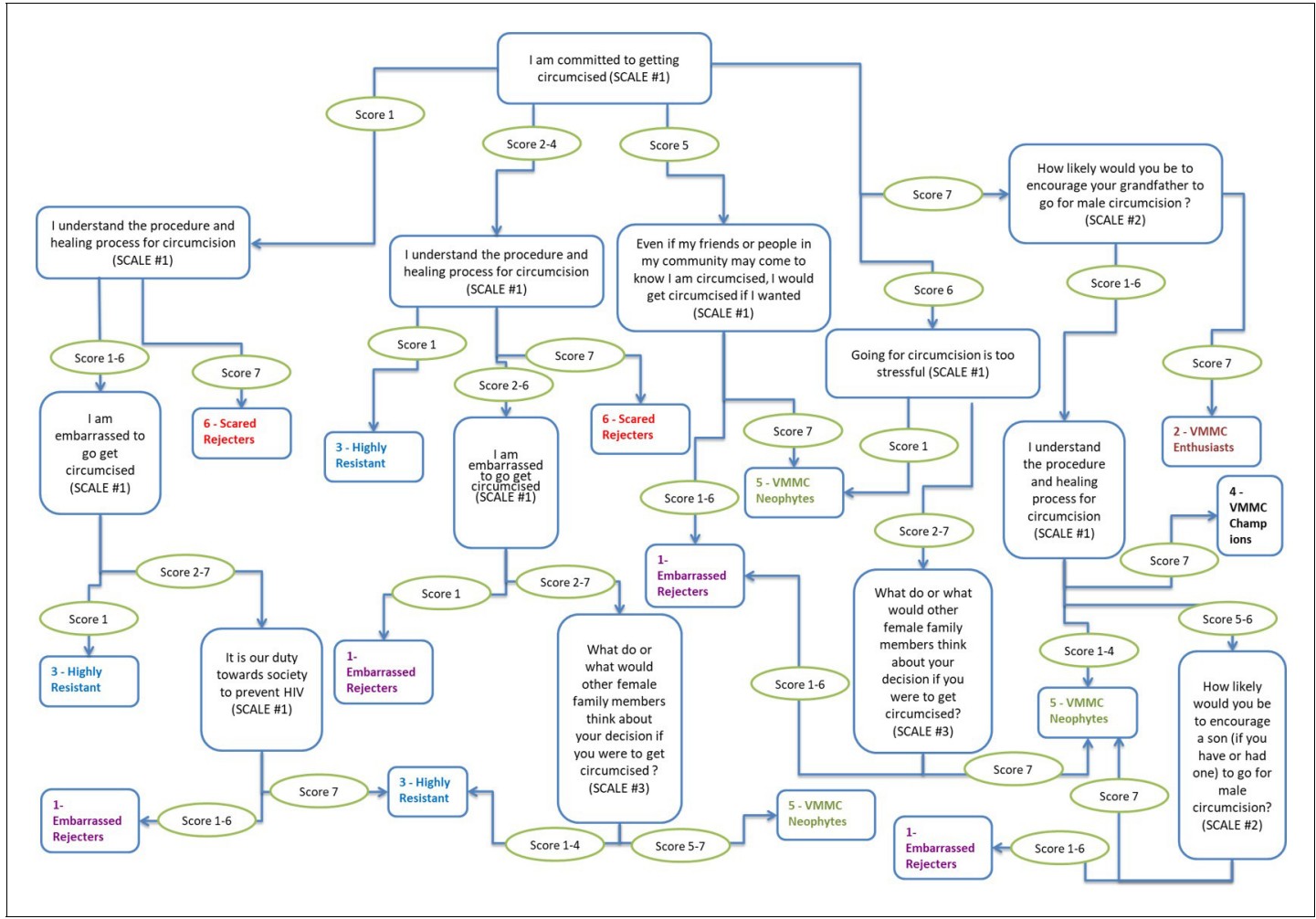

**Figure 4.** Segment typing tool-decision tree for Zimbabwe. Scale #1 (7-point scale): 7 = 'Strongly agree'; 4 = 'Neither agree nor disagree'; 1 = 'Strongly disagree'. Scale #2 (7-point scale): 7 = 'Would definitely encourage'; 4 = 'Would neither encourage nor discourage'; 1 = 'Would definitely NOT encourage'. Scale #3 (7-point scale): 7 = 'They think I definitely should get circumcised'; 4 = 'They don't have any particular opinion'; 1 = 'They think I definitely should NOT get circumcised'.

DOI: https://doi.org/10.7554/eLife.25923.011

The following source data is available for figure 4:

**Source data 1.** Segment typing tool questions.
DOI: https://doi.org/10.7554/eLife.25923.012

g., beliefs, fears, social influences, important for influencing them to take action to getting circumcised.

It is worthwhile to compare this behavioral-psychographic segmentation approach to a popular categorization provided by the Diffusion of Innovation Theory (*Rogers, 2003*). This framework specifies that in a given population, a novel product or innovation is first taken up by Innovators. It then spreads to Early Adopters, Late Adopters, and finally - after reaching critical mass - to Laggards. This typically happens in a sigmoidal ('S-shaped') fashion (*Dearing, 2009*). Diffusion of Innovations Theory emphasizes that different segments can be targeted in different ways. For instance, Innovators can be partners along the way to generate support for a new idea. In contrast, early Adopters are a testing ground and need close support, but Late Adopters are more likely to follow social norms and the fear of being left behind. This classification system is useful if behavioral-psychographic segmentation is not possible, but does not take the specific beliefs, attitudes, and emotions into account that place individuals in either of these categories. Without knowing the underlying reasons for why someone is an adopter or not, little can be done to anticipate and shift behaviors in a

specific case. In sum, while Diffusion of Innovations Theory provides a generalized and parsimonious classification, the segmentation approach we outline here provides a generalizable *method*. The segments that result from the behavioral-psychographic technique will be situation-specific, and therefore provide a foundation for more targeted interventions, than the fixed segments of the Diffusion of Innovations Theory. For instance, Diffusion of Innovations Theory outlines that Laggards need a great amount of familiarity with an innovation before they use it (*Rogers, 2003*). However, the characteristics of some segments we found that would most closely fit into the 'Laggards' category, such Embarrassed Rejecters and Highly Resistants in Zimbabwe, suggest that lack of familiarity is not at their root of resistance to VMMC.

The segment classification algorithms provide the ability to accurately classify any man in the field into his segment in order to provide messaging and interventions most appropriate to influencing him individually to take action to getting circumcised. For example, interpersonal communicators can use the segment classification tool to identify to which segment a man belongs and use a pre-scripted and/or ad hoc approach to counseling that man targeting the issues specific to that segment. For instance, if a man in Zimbabwe is identified as uncircumcised and belonging to the Scared Rejecters segment, the focus of the counseling for that man can quickly address the need to provide clarity on the process of getting circumcised, offer reassurance around the pain that will be felt (for example, by providing anchors about how much and when in the process of the procedure and healing period pain can be expected) and how the pain will be able to be managed. In addition, well-crafted mass media communications can be used, which focus on specific critical and differentiating factors for different segments. Men in the segments targeted will then self-selectively pay attention to those communications based on the interest the segmentation results have identified they have for them (for example, a Scared Rejecter will be more likely to respond to reassurance about pain than would an Embarrassed Rejecter).

Adding two additional questions to the end of the segment classification tool asking the self-reported number of times a man has sexual intercourse in a typical month and the self-reported number of different partners with whom a man has sex in a typical month would provide the ability to specifically identify and counsel men with regard to their personal estimated risk of HIV infection.

Having identified the relative proportions of each segment still uncircumcised, and the proportions of these men committed to getting circumcised or not, the segments in Zambia and Zimbabwe each have been prioritized for targeting. Highest-priority segments can be those that have the most 'low-hanging fruit' – uncircumcised men already mentally committed to getting circumcised (*Table 3*). Prioritizing these 'low-hanging fruit' is especially key in resource-constrained settings, where mass targeting can be replaced with smaller-scale, but focused, communications and interventions which specifically focus on the factors determining action for each targeted segment. Even mass media communication campaigns can be designed to create self-selected attention to them by target segments by using messaging especially relevant for each target segment. For example, a sports car commercial and a truck commercial will draw variable attention by those more interested in owning and driving each type of vehicle. Additionally, specific strategies for targeting each segment, based on the factors which can most influence men in that segment to take action, have been summarized. These strategies include key messages to use, appropriate use of mass media and interpersonal communicators, use of the men as advocates for VMMC, and perhaps the potential offering of device options for getting circumcised.

In Zimbabwe, for example, the highest priority segment for targeting is the Enthusiasts, who represent 21% of uncircumcised men, with 85% already committed to getting circumcised. They have relatively higher-risk sexual behavior, but are more likely to advocate for VMMC to other men, once circumcised, because of their strong beliefs about the benefits of VMMC for themselves and their community. For greatest influence, messaging for this segment should focus on detailed information on the procedure, the healing process and pain management to reduce uncertainty, and the potential for improved relationship with their partner, but they should also be counseled on their risky behavior. These communications would be best accomplished through interpersonal communicators. Once circumcised, these men should be actively engaged as VMMC advocates to promote it to other men of their peer age groups.

The relative sizes of segments within provinces differ at the province level, and are likely to differ at the district level. Understanding these differences provides programs with the ability to develop more localized programming. For example, districts with higher proportions of segments of men

**Table 3.** Segment targeting recommendations

| Country | Targeting priority | Rationale for targeting priority | Key messages | Use of mass media | Use of IPCs | Use of advocates |
|---|---|---|---|---|---|---|
| Zimbabwe | Enthusiasts | Large potential (21% of uncircumcised men) with 85% of segment committed; high risk behavior, but likely to advocate | Detailed info on procedure and healing process; pain management; improved relationship with partner | Not a target | Clarify pain during healing, time off work/school; counsel on potential increase in promiscuity | Engage as advocates |
| | Champions | Low potential (6% of uncircumcised men), but easy conversion (92% committed) and highly likely to advocate | Address uncertainty on healing process and pain during healing and procedure | Not a target | Address uncertainty on healing and pain; identify a friend-advocate to go with them for the VMMC | Engage as advocates |
| | Neophytes | Large potential (19% of uncircumcised men), and 57% committed; knowledge needed to inform commitment for rest | Full info on benefits and risks; clarify safe, skill of surgeon, healing process; where to get info and service | Personalize benefits, pain – how to manage it, accomplishment | Communicate full info on benefits, risks, safety, procedure and healing process | Use advocates to allay fears, share process, accompany |
| | Embarrassed Rejecters | Moderate potential (16% of uncircumcised men) but commitment low (22%) and embarrassment, fears high | VMMC becoming norm – be part of it; VMMC + condom use benefit; safe; how to manage pain, abstinence | VMMC norm, where service, reality of pain and how to manage it | VMMC norm, how to manage abstinence, reasons for pride, address myths believed | Provide community network of advocate support – VMMC as social norm |
| | Scared Rejecters | Moderate potential (17% of uncircumcised men) but commitment very low and fears/dissonance are strong | Safe procedure, low risk of complications; pain mgmt. during healing; improved relationship with partner | Not a target | Safe, low risk, expert service, pain real but manageable, involve partners | Use advocates to allay fears, share experience, accompany |
| | Highly Resistant | Large potential (21% of uncircumcised men), but hard to crack; knowledgeable, little fear; don't recognize need despite high-risk behavior | VMMC becoming social norm; address safety, service quality, privacy; pain management | Not a target | Acceptance of VMMC by wider community and advocacy from leaders; address fears with full info | Need advocates, communicating pride in VMMC and allaying fears |

| Country | Targeting priority | Rationale for targeting priority | Key messages | Use of mass media | Use of IPCs | Use of advocates |
|---|---|---|---|---|---|---|
| Zambia | Socially-supported Believers | Moderate potential (11% of uncircumcised men), high commitment (80%); likely advocate to broad audience, but dissonance | Address uncertainty on healing process and pain during procedure | Not a target | Address uncertainty on healing and pain; identify non-circumcised friends to go together for VMMC | Engage as advocates |
| | Self-reliant Believers | Moderate potential (9% of uncircumcised men), high commitment (86%), easy conversion to action; likely advocate for friends | Address questions about pain during procedure and healing process | Not a target | Address uncertainty on healing and pain; identify non-circumcised friends to go together for VMMC | Engage as advocates |
| | Knowledgeable Hesitants | Moderate potential (10% of uncircumcised men) and commitment (58%); key concern is safety | Protection benefits, VMMC + condom use benefit; safety, low risk of complications; pain management | Not a target | Safety - low risk (esp. for sexual life); pain is real, but manageable; expert service; involve partners | Use advocates to allay fears, share experience |
| | Friends-Driven Hesitants | Large potential (19% of uncircumcised men); 71% of segment committed; need add'l assurance, but relatively easily converted | VMMC social norm – be part of it; emotional benefits; detailed info on procedure and healing process | Personalized benefits; sense of accomplishment; low risk | VMMC norm, manage healing, clarify pain, service quality, availability; reasons for pride | Provide community network of support – social norm |
| | Scared Rejecters | Large potential (17% of uncircumcised men) but low commitment (37%) and strong concerns - fears and embarrassment | Safety, low risk of complications; pain management during healing; emphasize protection benefits | Not a target | Safety – credible info on low risk, expert service; pain is real, but manageable | Use advocates to allay fears, share experience, accompany them |

*Table 3 continued on next page*

*Table 3 continued*

| Country | Targeting priority | Rationale for targeting priority | Key messages | Use of mass media | Use of IPCs | Use of advocates |
|---|---|---|---|---|---|---|
| Zambia | Indifferent Rejecters | Large potential (27% of uncircumcised men), low commitment (21%); hard to crack; absence of motivation, while almost no concerns or fears | Full benefits and risks, HIV/STIs protection; clarify safety; address myths believed; where to get info and service | Full benefits, process, reality of pain and how to manage it; sense of accomplishment | Full benefits and risks; acceptance of VMMC by wider community and advocacy from leaders; dispel myths | Need a lot of advocates around, communicating pride in VMMC and allay fears |
| | Traditional Believers | Small potential (6% of uncircumcised men), but no need for support; high commitment (84%) driven by tradition - will get MC | Information on benefits and risks; where to get info and service | Not a target | Will benefit from short communication on benefits and risks; info on service/clinics | Not applicable |

DOI: https://doi.org/10.7554/eLife.25923.013

who are more fearful of surgery can prioritize particular interventions designed for these men, such as communications about the safety of the procedure and pain management during the procedure.

Targeted strategies using the segmentation results will require adoption of these results and recommended strategies into local VMMC program planning and tactical implementation efforts. While this will initially require additional efforts and potential reprogramming of some strategies, the potential to target and convert men to action for VMMC could increase demand, while doing so with greater efficiency. Similar segmentation solutions could be created and used in all countries where VMMC targets are far from achievement. To date, the segmentation tool has been implemented in pilot tests in both Zimbabwe and Zambia. This ongoing work is conducted in partnership with two non-governmental organizations with extensive experience in VMMC program delivery, Population Services International in Zimbabwe and Society for Family Health in Zambia. An evaluation of initial data is currently underway. Unpublished preliminary findings suggest that training community health workers to use the segmentation tool has measurable effects on overall conversion rates to the VMMC procedure in both countries. Policy makers in both Zimbabwe and Zambia have expressed interest in scaling up the segmentation approach once pilot data has been published.

The study has several limitations. The segmentation study was conducted with males aged 15–29 (core target audience), although the VMMC program is targeted towards men up to the age of 49. Thus, accuracy of the results among the older men requires additional evidence. HIV positive men also were out of scope of this research. For this research, data was collected only once and changes in the population over time will be measured in each country after implementation of the strategies in order to monitor the validity of the results. Any self-report design will also be subject to potential biases, such as social desirability bias. For instance, men may be reluctant to disclose unflattering attitudes to researchers. While self-report biases cannot be fully avoided, they are also pervasive among respondents in all contexts (*Nederhof, 1985*), and so unlikely to differ starkly between segments. In addition, the economic decision game asked men to select the option they thought other men would choose (rather than the one they themselves would select), thereby reducing perceived judgment on their own attitudes. Further studies could estimate the extent of existing biases by comparing face-to-face self-report with self-administered, or forced-choice designs.

Besides the limitations of the survey, there are some challenges related to translation of these results into action and their proper implementation. The crucial factors in regard to implementation of the results include proper prioritization of the segments, assessment of investments needed for implementation of interventions and strategies, created based on the results and effectiveness of interventions, and strategies need to be evaluated to ensure that they resonate with chosen segments.

In conclusion, behavioral-psychographic segmentation is a viable method to identify the diversity of drivers or barriers to a behavior that may exist within a group of healthcare beneficiaries. In this case study, we focused on revealing the different factors that prevented men in Zimbabwe and Zambia from taking up a crucial HIV prevention intervention. In the field, respondents can be allocated to segments with substantial accuracy, using simple decision trees. On the policy-making level, we then provided strategies for targeting the different segments with different messages and channels.

In any segmentation application, the identification of 'low-hanging fruit' segments will be crucial to maximize the impact of an intervention. Future public health strategies should therefore heed both the diversity of messaging and channels required to target different segments, but also consider prioritizing some segments over others depending on the likely impact and ease of conversion. If interventions are targeted to resonate with segments found through accurate field-based typing, HIV transmission rates in Zimbabwe and Zambia could decrease as the spread of HIV in men is reduced. Beyond the HIV application introduced here, behavioral-psychographic segmentation is likely to be a valuable tool whenever a group of stakeholders is diversified in their beliefs, emotions, and attitudes towards a target behavior.

## Materials and methods

### Instrumentation and data collection

Responses to a questionnaire, which formed the basis of behavioral segmentation, were collected in 2015 via face-to-face personal interviews among men in Zambia and Zimbabwe using structured quantitative surveys programmed on mobile devices. Surveys were conducted by male, local interviewers who were contracted by the market research company Ipsos in Zambia, and by Ipsos subcontractors in Zimbabwe. The design of the questionnaire utilized a framework based on the Integrated Behavior Model (IBM) (*Montano and Kasprzyk, 1990*; *Yzer, 2012*) and was guided by results from a qualitative stage of the research program, which indicated interest and information-seeking about VMMC, uncertainty about the need for VMMC and anxiety felt by a man about getting circumcised were key indicators of cognitive dissonance experienced by men as the main barrier to taking action to get circumcised (Eletskaya M et al., in preparation). Qualitative data was generated from two sources: journey mapping, and a decision-making game with subsequent hot-state interviews. From journey mapping, we obtained the temporal milestones in the process towards making a decision, and the proportion of men at each milestone. This method also uncovered the beliefs and attitudes for and against circumcision, as well as communication channels, that were relevant to men at each temporal stage. For example, mass communication was more relevant to men in earlier stages, whereas friends gained influence in later stages, and healthcare providers were most influential in very late stages of the decision-making process (Eletskaya M et al., in preparation). The decision-making game consisted of scenarios that simulated the real-world contexts of the participants. Men were presented with several hypothetical options (decisions) in response to a scenario, and were asked to select the one they thought a majority of other men would choose (Eletskaya M et al., in preparation). This was done in order to reduce the men's pressure towards carefully-deliberated and socially-desirable answers. Through the game, and subsequent hot-state interviews, we obtained additional information about beliefs, emotions, biases, and contextual factors, as well as triggers to act to get circumcised. As an example of a contextual factor, close male friends were found to be more influential on men's beliefs and attitudes than female partners (Eletskaya M et al., in preparation). Qualitative data was collected for a variety of strata of men, whether they were already circumcised or not; or if not, whether they already intended to undergo the procedure or not. This was done in order to obtain a broad picture of prevalent beliefs, biases, emotions, and contextual factors. The qualitative data were then used to inform which beliefs, emotions, attitudes, and contextual factors the survey questions should address. In turn, the survey formed the basis of the key differentiating variables for quantitative segmentation. Montaño used the IBM to quantitatively identify key beliefs about male circumcision in Zimbabwe (*Giles et al., 2005*; *Rhodes et al., 2007*; *Montaño et al., 2014*). Based on the IBM, intention is a key driver of behavior. Intention is driven by an individual's beliefs and perceived norms about the behavior and self-perceived control over being able to act on the behavior (*Fishbein and Ajzen, 2010*). Many of the key circumcision-related beliefs identified by Montaño in Zimbabwe were also identified as relevant beliefs in Zambia in other studies (*Price et al., 2014*); thus, the model was viewed as applicable for both countries. For the current research, the IBM was employed and modified to identify specific groups of men, differentiated from each other in terms of constructs underlying behavior. The beliefs identified by Montaño as influencing motivation for getting circumcised were used to inform and build these constructs. The core constructs that lead to an intention to get circumcised include attitudes about the act of getting circumcised (defined by both emotional feelings and functional beliefs about the behavior),

perceived norms (defined by both beliefs about others' expectations and others' behaviors) and personal agency to get circumcised (defined by both beliefs about personal control and personal efficacy). These determine intent, but other factors around perceptions of HIV risk and ability to control risk of infection, as well as structural factors such as availability of information and service for MC and service quality, are either barriers to or facilitators for taking-action. The qualitative phase of this research program also revealed that in some men, the absence of action in going for VMMC can be driven by additional motivational barriers rooted in cognitive dissonance (Eletskaya M et al., in preparation). Thus, measurement of the presence of cognitive dissonance (through 3 key components identified as important by the prior integrated qualitative study: interest and information-seeking about VMMC, uncertainty about the need for VMMC and anxiety felt by a man about getting circumcised) was implemented and combined with measurement of IBM theory constructs to provide a single design framework (*Figure 1*).

In the structured questionnaire, each construct was measured through a presentation of a number of questions for which respondents were asked to give an answer using a 7-point rating scale. The questionnaire was specifically designed to differentiate among men in their answers to the questions regarding their needs, perceptions, attitudes, beliefs and behaviors toward VMMC such that subsequent data analysis could partition the men into distinct segments, so that similarities are maximized within each segment and dissimilarities are maximized between segments. A categorical variable was used to determine where in the process men were in getting circumcised: (1) not aware of male circumcision as method for HIV prevention, (2) aware, but do not believe in benefits of male circumcision', (3) believe in the benefits, but not yet committed to getting circumcised, (4) committed to getting circumcised, but not yet scheduled it, (5) scheduled it, but not yet circumcised, (6) circumcised, but not advocating to other men to get circumcised, (7) circumcised and advocating to other men to get circumcised.

The sample consisted of 4001 men (both circumcised and uncircumcised), aged 15–29 years: 2001 men in Zimbabwe and 2000 in Zambia. Circumcised men were also included to be able to identify the full set of factors that lead to the actual decisions of seeking VMMC (not only intention to go for circumcision) and factors that influence attitudes and behaviors post-VMMC. For practical fieldwork cost and logistics purposes, the research targeted the districts with the highest concentrations of uncircumcised men in each country, cumulatively accounting for 80% of the uncircumcised populations in each country. Districts were first sorted from high to low by their populations of uncircumcised men. Then, the cumulative percentage of all uncircumcised men was calculated. Around 50% of districts were below the 80% cut-off point, such that the research was carried out in 38 of 72 districts in Zambia, and 35 of 61 districts in Zimbabwe. Country-level sample sizes (n = 2000 or 2001 men in each country) were determined based on experience with cluster segment sizes and the need for minimum sample size in the smallest resulting segment to be large enough for significance testing for differences across segments. Typically, cluster segmentations on consumers result in 4 to 8 segments, with the smallest segment representing as low as 5–10% of the total sample. The minimum desired sample for significance testing was judged as n = 100. Consequently, if this sample represents the smallest segment with a size of 5% of the total sample, the resulting total sample size should be n = 2000 (n = 100 * 20). Samples were distributed by age in proportion to the population size for each age group in each district. Households were randomly sampled in the selected districts and a male was approached in each household for the interview. If more than one eligible male lived in the household, selection among these males was made by random selection using a table of random numbers (*Kish, 1949*). Once a quota for an age group was reached in a district, only males who met open quota criteria were interviewed. If the household's selected male was not available or ineligible, the next household was approached. A small incentive was provided to compensate respondents for their time and refusal rates were low (<5%) and mostly due to men not having time to participate – consistent with rates experienced for other similar research. All respondent data was used in analysis of the results; none were excluded.

## Analytical approach
### Identification of segments
The analysis process was multi-stage, starting with reduction of the number of variables to a more manageable set using canonical correlations analysis (*Sheth and Johansson, 2011*; *Schnaars and*

*Schiffman, 1984*). Canonical correlation analysis showed that from the set of canonical correlation pairs (roots), the null hypothesis that correlations were zero could be rejected for some roots, for example the first 16 of 32 roots in the Zambia data (p=0.001 for roots 1–14, p=0.002 for root 15, p=0.024 for root 16; all subsequent roots p>0.1). For each of the canonical correlation pairs, eigenvalues were also computed. For instance, the first root had an eigenvalue of >1 in the Zambia data. The cumulative variance explained was also used to determine the optimal set of roots. The analysis identified 5 roots, explaining 68% of variance in the data in Zimbabwe and 7 roots, explaining 67% of variance in the data in Zambia, on which segmentations were based. At the second stage, hierarchical clustering was employed to explore possible partitions (e.g., 5 or 6 or 7 segments) and the results were used to identify the most appropriate number of segments (*Cross, 2008*). There is no unambiguous answer to the question which number of clusters is optimal, however standard and best practice approach is to find the solution which ensures that clusters are homogeneous within the clusters and maximally different between the clusters. Several statistics were referenced to determine whether the solution meets these criteria or not: Pseudo-F statistic, Pseudo-T statistic, and the overall $R^2$ (*Cross, 2008*). As an example, the Pseudo-F statistic gives the ratio of between-cluster variance to the variance within clusters, and when the index is plotted against the number of clusters, peaks in the graph indicate greater cluster separation. The Pseudo-T statistic is a measure of the difference between clusters merged together, and jumps in the index plotted against the number of clusters therefore point to an optimal number of clusters. After this stage, the cluster centers from this solution were refined with K-means clusters. At this stage an optimal solution was identified based on distances between clusters of >1.5, ensuring their distinctiveness (*Sheth and Johansson, 2011*; *Schnaars and Schiffman, 1984*; *Cross, 2008*). Then, clusters were profiled on the above-mentioned constructs in order to demonstrate which attitudes, beliefs, norms and behaviors characterize each segment.

Several statistically plausible solutions for the number of clusters were tested, and the final number of clusters was chosen from the possible solutions based on how practical and actionable the resulting segments were. The segment solutions were profiled by the number of variables, such as attitude and needs, and the likelihood to go through the procedure, as well as a number of socio-demographic variables. This enabled us to evaluate segmentations from a practical point of view, i.e. whether the segments were different enough on the variables that matter, and whether they made sense in terms of who to target, through which channels, and with what messages.

All statistical analysis was performed using SAS (RRID:SCR_008567) and SPSS (RRID:SCR_002865).

## Assessment of risk for HIV among segments

Two criteria on which the segments were profiled were 1) an estimation of relative risk for HIV infection based on self-reporting of behaviors, and 2) self-perceived risk of acquiring HIV and/or sexually transmitted infections (STIs). Indices were created for each of the two by creating a composite score using multiple questions measuring estimated relative risk and self-perceived risk of acquiring HIV, respectively.

The composite score for the estimated relative risk of HIV infection consisted of the self-reported number of times a man has sexual intercourse in a typical month and the self-reported number of different partners with whom a man has sex in a typical month. Other questions, such as contraction of STIs in the past, use of male condoms during the last sexual intercourse and anal sex engagement were eliminated from use in the index after initial analysis qualitatively identified them as less reliable (high level of socially desirable responses) and not contributing to differentiation between segments. Questions related to HIV infection risk, but not through sexual transmission (e.g. sharing personal objects or equipment such as injections, syringes or needles) also were excluded from the analysis because being circumcised would not mitigate those risks. Responses for the two questions used in the index were standardized to ensure that both questions contribute equally to the index score, and they were summed to create the new composite variable. The dataset of all scores of the composite variable was divided into three equal-size groups to designate low, moderate and high estimated risk for HIV infection based on sexual behaviors.

To assess self-perceived risk of HIV infection, respondents were asked to rate their level of agreement with two statements, using a 7-point Likert scale, where 1 meant 'Fully Disagree' and 7 meant

'Fully Agree'. The statements referenced respondents' belief about their personal likelihood of contracting HIV and STIs. The Index assessing perceived relative risk of HIV infection was created using the following simple algorithm: respondents rating 6, or 7 on the 7-point Likert scale for one or the other of the two statements were designated as High self-perceived risk, respondents rating both statements as 1 or 2 were designated as Low self-perceived risk. All remaining respondents were designated as Moderate self-perceived risk.

For each segment of men, the frequencies of that segment in the Low, Moderate and High estimated risk categories were contrasted with the frequencies for that segment of men across the Low, Moderate and High self-perceived risk categories.

## Segment typing algorithm

The chi-squared automatic interaction detection (CHAID) algorithm (*Miller et al., 2014*), which builds a decision tree of merging variables, was employed to identify key questions to ask men in the field. Ultimately, the typing tool needed to be practicable and suitable for use with just pen and paper. Accuracy was defined by a statistically significant cross-tabulation between the segments that were predicted by applying the algorithm to the sample, and the actual segments that were derived for the sample. Overall, it was possible to confidently predict in which segment any given man belonged (with >60% accuracy). Accuracy varied for different segments, and ranged from 39% to 78% in Zambia (mean = 61%, standard deviation = 13.1), and from 54% to 84% in Zimbabwe (mean = 71%, standard deviation = 10.4).

## Data availability

This study used data obtained from human participants. The dataset (anonymized survey responses) is owned by the governments of Zimbabwe and Zambia, and the authors have requested the respective governments to make the data publicly available. This request is currently subject to government approval. Until the data are publicly available, the data are made available upon reasonable request (criteria for access may apply subject to assessment by the respective governments). Requests for access to the data can be made to the following:

Zimbabwe
Ministry of Health and Childcare
Box CY1122, Causeway, Harare, Zimbabwe
Tel:+263 4 290 1210
Zambia
Ministry of Community Development, Mother and Child Health
Community House, Sadzu Road, Lusaka, Zambia
Tel:+260 211 225 327

## Acknowledgements

This work was supported with leadership by the Ministry of Health and Child Care (MoHCC) in Zimbabwe and the Ministry of Community Development, Mother and Child Health (MCDMCH) in Zambia. We thank our partners Population Services International in Zimbabwe and Society for Family Health in Zambia for collaborating on the implementation of this research.

## Additional information

### Funding

| Funder | Grant reference number | Author |
| --- | --- | --- |
| Bill and Melinda Gates Foundation | Contract #24210 | Steve Kretschmer |

The funders were involved in the study design, but had no role in data collection and interpretation, or the decision to submit the work for publication.

## Author contributions

Sema K Sgaier, Sehlulekile Gumede-Moyo, Steve Kretschmer, Conceptualization, Resources, Data curation, Software, Formal analysis, Supervision, Funding acquisition, Validation, Investigation, Visualization, Methodology, Writing—original draft, Project administration, Writing—review and editing; Maria Eletskaya, Data curation, Formal analysis, Supervision, Investigation, Visualization, Methodology, Writing—original draft, Writing—review and editing; Elisabeth Engl, Formal analysis, Validation, Visualization, Methodology, Writing—review and editing; Owen Mugurungi, Bushimbwa Tambatamba, Resources, Investigation, Project administration, Writing—review and editing; Gertrude Ncube, Sinokuthemba Xaba, Conceptualization, Resources, Supervision, Investigation, Project administration, Writing—review and editing; Alice Nanga, Resources, Formal analysis, Supervision, Investigation, Methodology, Project administration, Writing—review and editing; Svetlana Gogolina, Data curation, Formal analysis, Investigation, Methodology, Writing—review and editing; Patrick Odawo, Conceptualization, Resources, Supervision, Funding acquisition, Investigation, Project administration, Writing—review and editing

## Author ORCIDs

Sema K Sgaier http://orcid.org/0000-0002-8311-2686

## Ethics

Human subjects: In Zambia, ethical approval was received by ERES CONVERGE IRB, Ref. No. 2014-Aug-008. In Zimbabwe, ethical approval was received by MRCZ, Ref. No. MRCZ/A/1884. Consent to publish was received by all authors and the governments of Zambia and Zimbabwe. Written consent was obtained by all respondents. For those respondents below the age of 18 years (minors), both parental consent and consent from the interviewee were received. Consent forms were signed by both parent/guardian and minor in these cases. No respondents were under the age of 13 years.

## Decision letter and Author response

Decision letter https://doi.org/10.7554/eLife.25923.015
Author response https://doi.org/10.7554/eLife.25923.016

## Additional files

### Supplementary files

• Transparent reporting form
DOI: https://doi.org/10.7554/eLife.25923.014

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
