## [Decision Letter]

Thank you for submitting your article "A psychographic-behavioral segmentation approach for targeted demand generation in voluntary medical male circumcision" for consideration by *eLife*. Your article has been reviewed by two peer reviewers, and the evaluation has been overseen by a Reviewing Editor and Prabhat Jha as the Senior Editor. The following individuals involved in review of your submission have agreed to reveal their identity: Eliza Govender (Reviewer #1).

The reviewers have discussed the reviews with one another and the Reviewing Editor has drafted this decision to help you prepare a revised submission. We hope you will be able to submit the revised version within two months.

Market segmentation is a standard approach used in the private sector in introducing new products and for understanding and creating product demand. This paper describes an adaptation of this approach for a public health application viz addressing the challenge of low uptake of voluntary medical male circumcision (VMMC) for HIV prevention in two African countries with a high burden of HIV. A good case is made for market segmentation for demand creation that creates a more nuanced approach to targeting efforts based on psychographic segments and fills an important gap in HIV prevention research to create demand for adopting of new innovations/strategies to lower risk of HIV infection. As focus on preventive and promotive health expands and resources shrink there is a growing need to ensure appropriate targeting of interventions or combinations thereof to the right population for highest impact in efficient ways. Additionally, there is a growing public health need to understand demand creation strategies to introduce new technologies and interventions to populations or groups who would benefit most.

This manuscript provides a good opportunity to do this and more but not in its current form. Overall, it needs structure, focus and substantive editing to build on its novelty in its public health application as well as in extending its relevance beyond VMMC or beyond the two countries where the project was undertaken.

Specifically:

1) The manuscript is very dense and has too much information in the wrong places – too much detail where not necessary and not enough detail in the mixed methods approach used. In addition to the text there are graphs and 8 tables. In covering multiple issues simultaneously it is not doing justice to the novelty of the market segmentation approach and the outcomes. Consider either cutting down text and tables for a more focused and concise manuscript or separating into multiple manuscripts. What is the key message – define and stay focused on that.

2) The title suggests greater generalizability than what was actually done viz in two African countries.

3) More detail is needed in the methodology section: Rationale for site selection, catchment population and who the sample represents, sample size calculation, differences between the study sites – epidemiologically or culturally, value add of the qualitative data –, role in design of the questionnaires and quantitative component. Details on where was it undertaken, who was included, how was it conducted, with whom are needed.

4) The consumer journey mapping and behavioural economics game is interesting but how did it help identify context; how did it take into account multiple factors within and beyond the control of the individual, how did it impact strata within and between countries?

5) No rationale for focus on 15-29 year age group is provided. Is it based on HIV epidemiological data? Was the HIV prevalence or incidence rates similar in both countries?

6) The Results section needs a demographic characteristics table that includes variables by country and combined that includes mean age and range; employment status, years of schooling, linguistic or cultural characteristics; perception of HIV risk, social acceptability of VMMC; etc.

7) Segmentation – more generalizable categorization needed e.g. based on the theory of diffusion consumers fall into four broad categories viz Innovators, Early and Late Adopters and Laggards regardless of geo-spatial location or product.

8) More details on the segment typing algorithm – How was accuracy defined for prediction and for >60% was there any differences by type? In the Discussion section, the authors should mention how self-reporting may have biased the segmentation.

9) A key recommendation is the need for more customized approaches. Authors should comment on how this could be achieved in resource-constrained settings.

10) Authors should comment under Discussion how device choice was influenced by differences in what is available by country?

11) Also under Discussion, authors should underscore that the segments are only representative of the ages 15-29 years.

12) The Discussion would benefit from a concluding paragraph that explains the public health importance of this approach both in reaching men who are missing in the HIV prevention response in Africa and how segmentation at the various levels and use of the decision tree contribute to reducing the risk of HIV infection in men and thereby impact current high HV transmission rates in Africa.

---

## [Author Response]

[…] This manuscript provides a good opportunity to do this and more but not in its current form. Overall, it needs structure, focus and substantive editing to build on its novelty in its public health application as well as in extending its relevance beyond VMMC or beyond the two countries where the project was undertaken.Specifically:1) The manuscript is very dense and has too much information in the wrong places – too much detail where not necessary and not enough detail in the mixed methods approach used. In addition to the text there are graphs and 8 tables. In covering multiple issues simultaneously it is not doing justice to the novelty of the market segmentation approach and the outcomes. Consider either cutting down text and tables for a more focused and concise manuscript or separating into multiple manuscripts. What is the key message – define and stay focused on that.

We thank the reviewers for their advice on sharpening and focusing the manuscript. We have taken the comments on board, and have removed sections detailing regional differences between segments, details on the demographic characteristics by segment (but now provide a general overview of demographic and cultural background, as requested in point 4), the role of devices, and some of the figures and tables relating to that information.

Specifically, we have:

· Removed: “We identified geographic differences (at province and district level) in the distribution and size of segments to enable sub-national targeting of demand generation strategies. Given the potential role of circumcision devices in generating demand for VMMC [Fram et al., 1999], we identified segments where a device-driven VMMC strategy could be effective.”

· Removed the section ‘Application of segmentation to VMMC procedure option data’, as this can be split into a further manuscript.

· Removed the section ‘Demographic characteristics of segments’, as this is not central to our main message of showing the application of a behavioral-psychographic segmentation approach. Figure 2—figure supplements 2 and 3 were also cut.

· Removed the section ‘Regional analysis of segments’, to further sharpen the focus on our main message. The corresponding Figure 2—figure supplement 4 was also cut.

· Removed the section ‘Role of devices in targeting segments’, which will form part of a separate manuscript as recommended. The corresponding Table 3 was also cut, as was the column in Table 4 referencing devices targeting different segments and further references to devices in the text.

· Removed Figure 5 as the example of the segment typing decision tree in Zambia. Instead, we now only show the Zimbabwean example (Figure 4), and the typing tool questions for both countries (Figure 4—figure supplement 1).

· Changed the layout of Figure 3 so it fits onto one page.

· Simplified Figure 4 by removing the scale descriptions from the figure, and instead describing them in the legend.

We kept the detailed description of the segments throughout the Results section, including differences in risk perception, as it is central to understanding the kind of data this method can yield.

We also added greater detail to the mixed methods approach as requested, as per the comments below.

2) The title suggests greater generalizability than what was actually done viz in two African countries.

The method we outline aims to be generalizable beyond this case study in two African countries. We have changed the title in order to qualify generalizability, and highlight that this is a case study, to: “A case study for a psychographic-behavioral segmentation approach for targeted demand generation in voluntary medical male circumcision”.

3) More detail is needed in the methodology section: Rationale for site selection, catchment population and who the sample represents, sample size calculation, differences between the study sites – epidemiologically or culturally, value add of the qualitative data –, role in design of the questionnaires and quantitative component. Details on where was it undertaken, who was included, how was it conducted, with whom are needed.

We appreciate the reviewers’ pointer to include more detail in the methodology section. These comments greatly helped us clarify what needed to be included. We have now added supplementary information as requested:

· Rationale for site selection:

We have now clarified this in the response to the bullet point “Differences between the study sites – epidemiologically or culturally”.

· Catchment population and who the sample represents:

The table requested for point 6 now provides more information about the catchment population. In the manuscript, we already outline who the sample represents: “Samples were distributed by age in proportion to the population size for each age group in each district. […] If the household’s selected male was not available or ineligible, the next household was approached.”

· Sample size calculation:

We now add our rationale for sample size calculation: “Country-level sample sizes (n=2,000 or 2,001 men in each country) were determined based on experience with cluster segment sizes and the need for minimum sample size in the smallest resulting segment to be large enough for significance testing for differences across segments. […] Consequently, if this sample represents the smallest segment with a size of 5% of the total sample, the resulting total sample size should be n=2,000 (n=100 * 20).”

· Differences between the study sites – epidemiologically or culturally:

In response to a request for information, no information was made available by the Zimbabwean and Zambian governments epidemiological differences at the district level. Culturally, we took into account that circumcision was already an established practice in some districts and medical circumcision rates varied across districts since establishment of the services in each country. For practical fieldwork purposes, we sought to field the research in the fewest total number of districts which represented a high proportion (set as 80%) of total uncircumcised men in each country. We then ranked the districts in each country by their populations of uncircumcised men, high to low and calculated the cumulative% of total uncircumcised population through the rankings. We then selected the top-ranked districts cumulatively accounting for 80% of uncircumcised men in each country in which to field the surveys. We now clarify this:

“For practical fieldwork cost and logistics purposes, the research targeted the districts with the highest concentrations of uncircumcised men in each country, cumulatively accounting for 80% of the uncircumcised populations in each country. […] Around 50% of districts were below the 80% cut-off point, such that the research was carried out in 38 of 72 districts in Zambia, and 35 of 61 districts in Zimbabwe.”

· Value-add of the qualitative data – role in design of the questionnaires and quantitative component:

Qualitative data was generated from two sources: journey mapping, and a decision-making game with subsequent hot-state interviews. These data were then used to inform development of the survey questions, which in turn formed the basis of the key differentiating variables for segmentation.

From journey mapping, we obtained:

- Temporal milestones in the process towards making a decision, and the proportion of men at each milestone.

- Beliefs and attitudes for and against circumcision that were relevant to men at each temporal stage.

- Roles of influencers in developing those beliefs among men, including a range from female partners and male friends to healthcare providers or media, e.g., TV billboards or radio.

From the decision-making game and subsequent hot-state interviews, we obtained:

- Additional information about beliefs and emotions, biases, and contextual factors (such as who influenced men).

- Triggers for men to act to get circumcised.

We have now clarified what the qualitative research added in the Materials and methods section:

“Qualitative data was generated from two sources: journey mapping, and a decision-making game with subsequent hot-state interviews. […] In turn, the survey formed the basis of the key differentiating variables for quantitative segmentation.”

· Details on where was it undertaken, who was included, how was it conducted, with whom?

In the last paragraph of the Materials and methods subsection “Instrumentation and data collection”, we specified who was included in the sample, and how the survey was conducted (first paragraph of the aforementioned subsection). We now add that: “surveys were conducted by male, local interviewers who were contracted by the market research company Ipsos in Zambia, and by Ipsos sub-contractors in Zimbabwe”.

4) The consumer journey mapping and behavioural economics game is interesting but how did it help identify context; how did it take into account multiple factors within and beyond the control of the individual, how did it impact strata within and between countries?

We now answer these points in response to point 3, where we elaborate on the qualitative methods used.

5) No rationale for focus on 15-29 year age group is provided. Is it based on HIV epidemiological data? Was the HIV prevalence or incidence rates similar in both countries?

We thank the reviewers for highlighting that this issue needs clarification. In the Introduction we provided a rationale for focusing on this age group: “This quantitative study was conducted among males 15-29 years old, given previous evidence that identified this as the most efficient and impactful age for the VMMC programs in both countries to target [Awad et al., 2015a; Awad et al., 2015b].”

We now add to this sentence to be clearer on the rationale provided in the source papers:

“In an age-structured mathematical model, Awad et al. [[Awad et al., 2015a; Awad et al., 2015b] assessed the impact of prioritizing different age groups for VMMC in Zimbabwe [[Awad et al., 2015a] and Zambia [[Awad et al., 2015b]. […] Therefore, we focused our study on this age group.”

HIV prevalence rates in Zimbabwe (14.7%) and Zambia (12.9%) are comparable (http://www.unaids.org/en/regionscountries/countries); however, there is regional variation beyond the scope of this article.

6) The Results section needs a demographic characteristics table that includes variables by country and combined that includes mean age and range; employment status, years of schooling, linguistic or cultural characteristics; perception of HIV risk, social acceptability of VMMC; etc.

We thank the reviewers for pointing out the need for this information. We now provide this data as Figure 1—figure supplement 1 (demographic and cultural background) and 2 (social acceptability of VMMC + perception ofHIV/STI risk). We collected these variables by country, and therefore present them in this way.

We added a ‘Population characteristics’ section as follows: “Figure 1—figure supplement 1 shows the demographic and cultural characteristics of the population sample in Zambia and Zimbabwe. […] The starkest difference was that in Zambia, a much greater share of the population was only educated to primary-school level, and a smaller percentage was employed than in Zimbabwe.”

As noted in the response to point 1, we have removed the sections and figures relating to ‘Demographic characteristics of segments’ and ‘Regional analysis of segments’, as one general comment was that too much peripheral information was provided in the paper. While the new figures requested in this point add a general picture of the characteristics of the studied population, the removed figures delved into the details of those characteristics by segment, and showed how they were distributed geographically.

7) Segmentation – more generalizable categorization needed e.g. based on the theory of diffusion consumers fall into four broad categories viz Innovators, Early and Late Adopters and Laggards regardless of geo-spatial location or product.

We thank the reviewers for highlighting the segments identified by the Diffusion of Innovation Theory. However, we think that one of the key advantages of the approach introduced here is that it does not rely on a generalizable, and therefore not targeted, categorization. Instead, if offers a generalizable method, which will result in situation-specific categorizations that reveal concrete drivers of behavior.

We have added a paragraph to the Discussion in which we introduce Diffusion of Innovation Theory and discuss the difference between the two approaches:

“It is worthwhile to compare this behavioral-psychographic segmentation approach to a popular categorization provided by the Diffusion of Innovation Theory [Rogers, 2003]. […] However, the characteristics of some segments we found that would most closely fit into the ‘Laggards’ category, such Embarrassed Rejecters and Highly Resistants in Zimbabwe, suggest that lack of familiarity is not at their root of resistance to VMMC.”

8) More details on the segment typing algorithm – How was accuracy defined for prediction and for >60% was there any differences by type?

We thank the reviewers for requesting this important clarification. We have now added the requested detail on the segment typing algorithm to the Materials and methods section, and amended the relevant section as follows:

“The chi-squared automatic interaction detection (CHAID) algorithm, which builds a decision tree of merging variables, was employed to identify key questions to ask men in the field. […] Accuracy varied for different segments, and ranged from 39% to 78% in Zambia (mean = 61%, standard deviation = 13.1), and from 54% to 84% in Zimbabwe (mean = 71%, standard deviation = 10.4).”

In the Discussion section, the authors should mention how self-reporting may have biased the segmentation.

This is indeed an issue to consider, and we have added the following paragraph to the Discussion:

“Any self-report design will also be subject to potential biases, such as social desirability bias. […] Further studies could estimate the extent of existing biases by comparing face-to-face self-report with self-administered, or forced-choice designs.”

9) A key recommendation is the need for more customized approaches. Authors should comment on how this could be achieved in resource-constrained settings.

We thank the reviewers for pointing out this aspect of implementation. Segmentation allows programs to focus on targets that are most likely to generate impact. Therefore, the technique is highly useful for resource-constrained settings. To clarify this point, we have added the following:

“Prioritizing these ‘low-hanging fruit’ is especially key in resource-constrained settings, where mass targeting can be replaced with smaller-scale, but focused, communications and interventions which specifically focus on the factors determining action for each targeted segment. […] For example, a sports car commercial and a truck commercial will draw variable attention by those more interested in owning and driving each type of vehicle.”

10) Authors should comment under Discussion how device choice was influenced by differences in what is available by country?

In line with overall recommendations, we have now sharpened the focus of our paper and have taken out the section on device choice.

11) Also under Discussion, authors should underscore that the segments are only representative of the ages 15-29 years.

We have added this qualifier in the Discussion (first paragraph).

12) The Discussion would benefit from a concluding paragraph that explains the public health importance of this approach both in reaching men who are missing in the HIV prevention response in Africa and how segmentation at the various levels and use of the decision tree contribute to reducing the risk of HIV infection in men and thereby impact current high HV transmission rates in Africa.

We thank the reviewers for this suggestion, and we have now added a concluding paragraph to the paper addressing these points:

“In conclusion, behavioral-psychographic segmentation is a viable method to identify the diversity of drivers or barriers to a behavior that may exist within a group of healthcare beneficiaries. […] Beyond the HIV application introduced here, behavioral-psychographic segmentation is likely to be a valuable tool whenever a group of stakeholders is diversified in their beliefs, emotions, and attitudes towards a target behavior.”